# Comparison of Conventional and Radiomic Features between ^18^F-FBPA PET/CT and PET/MR

**DOI:** 10.3390/biom11111659

**Published:** 2021-11-09

**Authors:** Chien-Yi Liao, Jun-Hsuang Jen, Yi-Wei Chen, Chien-Ying Li, Ling-Wei Wang, Ren-Shyan Liu, Wen-Sheng Huang, Chia-Feng Lu

**Affiliations:** 1Department of Biomedical Imaging and Radiological Sciences, National Yang Ming Chiao Tung University, Taipei 11221, Taiwan; wl0315178315875@ym.edu.tw (C.-Y.L.); s10604004@ym.edu.tw (J.-H.J.); 2Department of Radiation Oncology, Taipei Veterans General Hospital, Taipei 11217, Taiwan; chenyw@vghtpe.gov.tw (Y.-W.C.); lwwang@vghtpe.gov.tw (L.-W.W.); 3Department of Nuclear Medicine, Taipei Veterans General Hospital, Taipei 11217, Taiwan; cylee3@vghtpe.gov.tw; 4Department of Nuclear Medicine, Cheng Hsin General Hospital, Taipei 11220, Taiwan; rsliu@vghtpe.gov.tw; 5Department of Nuclear Medicine, Taipei Medical University Hospital, Taipei 11031, Taiwan

**Keywords:** ^18^F-FBPA, boron neutron capture therapy (BNCT), malignant tumor, radiomics, positron emission tomography, PET/MR, PET/CT, T/N ratio, nuclear medicine

## Abstract

Boron-10-containing positron emission tomography (PET) radio-tracer, ^18^F-FBPA, has been used to evaluate the feasibility and treatment outcomes of Boron neutron capture therapy (BNCT). The clinical use of PET/MR is increasing and reveals its benefit in certain applications. However, the PET/CT is still the most widely used modality for daily PET practice due to its high quantitative accuracy and relatively low cost. Considering the different attenuation correction maps between PET/CT and PET/MR, comparison of derived image features from these two modalities is critical to identify quantitative imaging biomarkers for diagnosis and prognosis. This study aimed to investigate the comparability of image features extracted from ^18^F-FBPA PET/CT and PET/MR. A total of 15 patients with malignant brain tumor who underwent ^18^F-FBPA examinations using both PET/CT and PET/MR on the same day were retrospectively analyzed. Overall, four conventional imaging characteristics and 449 radiomic features were calculated from PET/CT and PET/MR, respectively. A linear regression model and intraclass correlation coefficient (ICC) were estimated to evaluate the comparability of derived features between two modalities. Features were classified into strong, moderate, and weak comparability based on coefficient of determination (**r^2^**) and ICC. All of the conventional features, 81.2% of histogram, 37.5% of geometry, 51.5% of texture, and 25% of wavelet-based features, showed strong comparability between PET/CT and PET/MR. With regard to the wavelet filtering, radiomic features without filtering (61.2%) or with low-pass filtering (59.2%) along three axes produced strong comparability between the two modalities. However, only 8.2% of the features with high-pass filtering showed strong comparability. The linear regression models were provided for the features with strong and moderate consensus to interchange the quantitative features between the PET/CT and the PET/MR. All of the conventional and 71% of the radiomic (mostly histogram and texture) features were sufficiently stable and could be interchanged between ^18^F-FBPA PET with different hybrid modalities using the proposed equations. Our findings suggested that the image features high interchangeability may facilitate future studies in comparing PET/CT and PET/MR.

## 1. Introduction

Boron neutron capture therapy (BNCT) is a type of targeted radiotherapy. BNCT shows promising results in treating lung cancer, recurrent head and neck cancer, sarcomas, and high grade brain tumors [1,2]. Due to the high local tumor control rate, BNCT is considered a promising treatment for malignant tumors [3]. Tumor cells tend to show a higher uptake level of the ^10^B-containing drug, such as Boronophenylalanine (BPA), than do normal tissues. The high tumor specificity of BPA is because of the selective transport by L-type amino acid transporter 1, which is upregulated in cancers [4]. Thermal neutrons are captured with high probability by BPA drugs, leading to the nuclear reaction of ^10^B (n, α, γ) ^7^Li. High linear energy transfer particles, i.e., alpha particle and ^7^Li, are generated in this reaction [5,6,7]. An effective BNCT for cancer treatment requires a sufficient tumor-to-normal tissue ratio (T/N ratio, greater than 2.5) of BPA [8]. A radio-labeled phenylalanine analogue for positron emission tomography (PET), 4–borono-2-^18^F-fluoro-phenylalanine (^18^F-FBPA), has been used to evaluate the T/N ratio of BPA in clinical practice [9,10,11]. The ^18^F-FBPA may be considered a superior tumor-specific tracer to the frequently-used PET tracer, 2-^18^fluoro-2-deoxy-D-glucose (^18^F-FDG) with regard to reducing false positive rates caused by uptakes in normal gray matter and inflammation tissues [4,12,13].

Due to high soft-tissue contrast, functional imaging capability and low radiation exposure, PET/MR becomes more available and partly replaces PET/CT in some clinical examinations including the brain, head and neck [14]. However, PET/CT still has advantages compared to PET/MR, such as good lung-to-bone contrast, accurate attenuation correction, and relatively low costs [14]. Furthermore, PET/CT technology has strong and long clinical experience and does not require MRI skills for nuclear physicians in comparison to PET/MR. Accordingly, both PET/CT and PET/MR play critical roles in investigating tumor metabolism. Several PET-derived characteristics are applied in the diagnosis of cancers, including conventional and radiomic features. Conventional features are mostly voxel-based to reflect the uptake of PET tracers in tumor cells, such as standardized uptake values (SUV) and the T/N ratio [15]. Radiomic features are high throughput and quantitative characteristics which describe the image pattern and heterogeneity of a tumor [16,17,18]. Nevertheless, variability of the image features is observed between PET/MR and PET/CT. This is potentially caused by the different reconstruction processes of images, such as the correction of attenuation and partial volume effect as well as the noise reduction [19,20].

The variations in values of image features extracted from different PET modalities have been discussed in a few previous studies [21,22]. However, these studies mainly focused on ^18^F-FDG PET images, and no investigation of ^18^F-FBPA in BNCT application was reported. Because ^18^F-FBPA imaging is critical in evaluating suitability and outcome with BNCT cancer management, comparing ^18^F-FBPA imaging features between two common PET modalities in practice was needed. The aim of this study was to investigate the comparability of conventional and radiomic features between ^18^F-FBPA PET/CT and PET/MR acquired from same patients. We further proposed equations to interchange image features with sufficient comparability between PET/CT and PET/MR modalities.

## 2. Materials and Methods

### 2.1. Patient Cohort

We retrospectively analyzed patients with malignant tumors who underwent both ^18^F-FBPA PET/CT (Discovery MI DR, GE Healthcare, Chicago, IL, USA) and PET/MR (SIGNA PET/MR, GE Healthcare, Chicago, IL, USA) on the same day. Fifteen patients were included with the confirmation of the following conditions: (1) pathological diagnosis of primary or recurring malignant tumors; (2) at least one detectable lesion identified in the PET image; and (3) acceptable PET image quality evaluated by nuclear physicians.

### 2.2. Preprocessing of PET Data

Two preprocessing steps of the PET images were applied to improve the reliability of radiomics analysis. First, image resolution adjustments were performed to re-sample all voxel sizes to 1.5 × 1.5 × 1.5 mm^3^ for each PET examination. Second, the SUV map was calculated to control the deviation of patients’ body weight and the decay of isotopes [23,24,25]. All steps of image pre-processing and feature extraction were performed by the previously published MR Radiomics Platform (MRP) [26,27] with extended functions to calculate SUV maps complied with the Image Biomarker Standardization Initiative (IBSI) [28]. Conventional features, including T/N ratio and the maximum, median and minimum values of SUV, were then derived from each PET dataset.

### 2.3. Radiomic Feature Extraction

A semi-automatic method based on the maximum SUV was used to define tumor region of interest (ROI). In this study, we defined the regions with SUV higher than 50% of SUV_max_ as the ROIs. This criteria for tumor segmentation is commonly used in clinical PET imaging [29]. The ROIs were delineated on both modalities of PET images, respectively, and reviewed by a team comprised of experienced nuclear medicine physicians and radiation oncologists for pre-BNCT evaluation. Figure 1 shows representative PET/CT and PET/MR images of patients with brain tumor and head and neck cancer. Finally, 449 radiomic features, including geometric, histogram, texture and wavelet, were extracted from each PET dataset. The histogram features described the intensity distribution of the ^18^F-FBPA uptake within ROI. Geometric features measured the 3D shape and size of tumor ROIs. Texture features quantified the heterogeneity of ROIs based on the gray level co-occurrence matrices (GLCM) and gray level run length matrix (GLRLM) [30,31,32]. Wavelet features provided the detail of images by filtering image signals based on different levels of spatial frequency. Wavelet decomposition was performed by applying low-(L) and high-(H) pass dimensional filters along three image axes, generating eight decomposed image sets: LLL, LLH, LHL, LHH, HLL, HLH, HHL and HHH filtered images. Histogram and texture features were calculated on the original images and each of the eight wavelet filtered images. The formulae for feature calculations are provided in Appendix A.

### 2.4. Statistical Analysis

Correlation between the ^18^F-FBPA radiomic features extracted from PET/CT and PET/MR was assessed using a linear regression model. The coefficient of determination (r^2^) between PET/CT and PET/MR was applied to evaluate the consistency of derived features [33]. The formulae for the interconversion of ^18^F-FBPA radiomic features between PET/CT and PET/MR were also derived based on the coefficients and intercepts of linear regression. The intraclass correlation coefficient (ICC) was calculated to assess the comparability of radiomic features. A two-way mixed single model of ICC were used for comparison of PET/CT and PET/MR image features [34,35]. In this study, the features with r^2^ larger than 0.85 and ICC larger than 0.75 were regarded as strong comparability; the features with r^2^ larger than 0.5 and ICC larger than 0.5 were regarded as moderate comparability; the features that didn’t match above two conditions were regarded as weak comparability between PET/CT and PET/MR [36,37]. Furthermore, a paired *t*-test was used to evaluate whether the r^2^ and ICC of wavelet features showed significant difference compared to the original images. The workflow of data analysis is shown in Figure 2.

## 3. Results

### 3.1. Clinical Characteristics of Patients

This retrospective study included 15 patients (12 brain cancers and 3 head and neck cancers). The averaged patient age at receiving treatment of BNCT was 55.4 years old (range: 13–88). Patient characteristics are summarized in Table 1. 

### 3.2. Linear Correlation between PET/CT and PET/MR Features

As shown in Figure 3a, all four conventional features expressed r^2^ were > 0.90. Furthermore, most of the histogram features were well fitted by the linear models with r^2^ values > 0.85. More than half of the geometric features presented r^2^ values < 0.85. For the wavelet and GLCM features, the distribution of r^2^ values exhibited lower medians (0.69 and 0.73, respectively) and wider variances, resulting in only 25%, and 45.6% showed r^2^ > 0.85, respectively. Figure 3b shows distribution of r^2^ for the radiomic features with different wavelet filtering (i.e., various combinations of low-pass or high-pass filters along three imaging axes). We found that the features based on high-pass wavelet filters might enhance the inhomogeneous compositions of PET/CT and PET/MR (such as the edges and fine details) and therefore reduced the comparability between the two modalities. For example, the features based on the HHH wavelet showed a significantly lower median of r^2^ compared to those without wavelet features (*p* < 0.001); whereas the r^2^ of LLL-based features didn’t show a significant difference compared to those without wavelet features (Figure 3b). 

### 3.3. Intraclass Correlation between PET/CT and PET/MR Features

Distribution of ICCs of each feature type is shown in Figure 3c. All four conventional features showed ICC values > 0.95. Histogram (87.5% of features) and GLRLM (82.0% of features) were highly correlated between PET/CT and PET/MR with ICC values > 0.75. On the other hand, only 59.1% of GLCM features and 59.7% of wavelet features exhibited ICC values > 0.75, indicating these two feature types were more dissimilar between PET/CT and PET/MR. The ICC distribution of each wavelet filter type is showed in Figure 3d. Similar to the results of linear correlation analysis (r^2^ distribution), the features based on HHH wavelet showed a significantly lower median of ICC values compared to those without wavelet features (*p* < 0.001); whereas the ICC values of LLL-based features didn’t show a significant difference compared to the none-filtered features (Figure 3d).

### 3.4. Features with High Comparability for Interchange between PET/CT and PET/MR

As shown in Figure 4a, 81.2% of histogram and 63.7% of GLRLM features reached the criteria of strong comparability (r^2^ > 0.85 and ICC > 0.75). Only 37.5% of geometric features reached strong comparability, and the rest presented moderate comparability. 45.6% of GLCM features and 24.2% of wavelet features presented strong comparability. Finally, all conventional features showed strong comparability between PET modalities. Figure 4b reveals percentages of each comparability level in different wavelet filters. A larger portion of features with strong comparability was found in none-filtered and LLL wavelet features compared to other wavelet types comprising high-pass filters. The HHH filter type had the least portion of features with strong comparability. The inconsistency of r^2^ and ICC value between two PET modalities in wavelet features was possibly caused by the high intrinsic variability of the applied wavelet filters. 

In total, 128 out of 449 radiomic features were considered to have strong comparability, and 191 out of 449 radiomic features were considered as having moderate comparability. We provided slope and intercept of linear equations for interchanging the 323 features (four conventional features with strong comparability; 128 radiomic features with strong comparability; 191 radiomic features with moderate comparability) between PET/CT and PET/MR in Appendix A. 

Figure 5 shows the linear regression of the representative features with high r^2^ and ICC values from each feature type. For instance, volume had the highest r^2^ and ICC value among geometric features (Figure 5c). The most upper right sample point of volume (Figure 5c), energy (Figure 5b), and run length non-uniformity (Figure 5e) appeared to be from the same patient. After removing this leverage point, the r^2^ values of volume, energy and run length non-uniformity slightly decreased to 0.93, 0.95, and 0.93, respectively. 

## 4. Discussion

BNCT provides a promising treatment option for patients with malignancies due to a proven ability to improve overall survival and local tumor control [38]. ^18^F-FBPA PET for pre-BNCT evaluation can be performed in either PET/CT or PET/MR modalities in practice. Both PET/CT and PET/MR have shown their own benefits in different situations of BNCT. Accordingly, this hybrid image modality has gradually been applied to evaluate brain tumors [39,40], especially the intractable high-grade brain tumor which is the primary indication of BNCT [41,42,43]. 

The T/N ratio of ^18^F-FBPA PET is an important indicator for considering BNCT. Previous clinical trials revealed that a T/N ratio > 2.5 was suitable for BNCT [44]. However, T/N ratios could be influenced by selections of normal tissues [45], resulting in inconsistent T/N ratios that might cause bias for further BNCT. Searching feasibly additional ^18^F-FBPA imaging features as the pre-BNCT indicators is therefore a high profile issue in BNCT research. PET radiomics have been applied in ^18^F-FDG PET for tumor staging, evaluating therapeutic efficacy, and predicting recurrence and metastasis [46]. This study explored the effect of PET modality on ^18^F-FBPA image features. Our findings could be beneficial for the development of ^18^F-FBPA radiomics as prognostic biomarkers. 

In this study, we categorized the comparability of ^18^F-FBPA image features between PET/CT and PET/MR into three levels: strong, moderate, and weak comparability. A strong comparability (r^2^ > 0.85 and ICC > 0.75) indicated that the features may not be influenced by PET modalities with regard to attenuation correction methods. These features exhibited slopes close to 1 and small intercepts in linear regression models. We therefore suggested that these features could be interchanged between PET/CT and PET/MR without adjustments. Features with moderate comparability (0.5 < r^2^ < 0.85 and 0.5 < ICC < 0.75) had certain amounts of differences between the PET/CT and PET/MR, but could be interchanged between different modalities based on the proposed linear transformation equations. Finally, radiomic features with low comparability (r^2^ < 0.5 or ICC < 0.5) may exhibit huge differences between two modalities, and therefore were not recommended to pool together or interchange. 

Several factors might cause variations of imaging features between PET/CT and PET/MR modalities. First, PET/CT and PET/MR performed the photon attenuation correction in accordance with different algorithms. Available data indicated that the attenuation correction for PET/CT and PET/MR showed an error of <10% for the SUV maximum (SUV_max_) and mean (SUV_me__an_) [47,48]. However, different attenuation correction approaches and anatomical images (CT vs. MR) may potentially cause variations of SUV maps reflecting on the image geometry and textural patterns. Our data showed that the texture features had a relatively high proportion of weak comparability features. Second, the partial volume correction based on anatomical images for PET appeared essential to avoid underestimation of SUVs [49]. Third, the decay of ^18^F-FBPA activity in vivo over time may cause differences in SUVs. In the current study, most PET/MR images were performed within 60 min after PET/CT. A previous study regarding the time course of ^18^F-FBPA uptake showed that the SUV_max_ could maintain a high level from 20 to 120 min after ^18^F-FBPA injection in most brain tumors [50]. Duration of ^18^F-FBPA uptake and decay, however, may still partially influence radiomic features. 

Among the histogram features, only skewness and uniformity features showed a weak comparability. The skewness measured the asymmetry of the distribution of SUVs with respect to the mean SUV, and the uniformity was a measure of the sum of the squares of SUV values within the tumor ROI. In this study, we defined the regions with SUV higher than 50% of SUV_max_ as the ROIs, which is a commonly used segmentation criteria in clinical PET imaging [29]. Even if the similar thresholds were applied, the segmentation might be still influenced by the utilized attenuation correction methods, resulting in a slight difference on ROI boundaries between PET modalities. We also observed that five (including compactness 1, compactness 2, spherical disproportion, maximum 3D diameter and sphericity) of eight geometric features presented a moderate comparability rather than a strong comparability. This might also result from the variation of the ROI due to different attenuation correction methods. In short, most of the histogram and geometric features could reach a moderate or a strong level of comparability between PET/CT and PET/MR. 

In this study, texture analysis included GLCM and GLRLM features. The GLCM-texture described the intensity variations between neighboring voxels, and the GLRLM-texture described the tendency of linear distribution for voxels with the same SUV. In comparison with GLCM-texture, more GLRLM-texture features presented a strong comparability between PET/CT and PET/MR. This may be because the distribution of neighboring voxels was more sensitive to the adjustment of attenuation correction or image reconstruction [51]. In addition, we evaluated the influence of wavelet filters for the comparability of radiomics between PET modalities. The use of high-pass filters could enhance the edges of the image and makes it appear sharper [52]. This may lead to poorer reproducibility and comparability of radiomics between different PET modalities. In summary, radiomic features associated with texture (measuring the fine detailed cross image distribution) and high-pass wavelet filtering may present a lower comparability between PET/CT and PET/MR.

Several limitations should be carefully considered in this study. First, the SUV map corrected for the decay of activity caused by the time delay between PET/CT and PET/MR in ^18^F-FBPA uptake. However, the potential pharmacokinetic effect of ^18^F-FBPA between two PET acquisitions on the image features may require further investigation. Second, this study included a relatively small sample size due to the limited number of clinical BNCT cases worldwide [53]. Nevertheless, ^18^F-FBPA imaging features still showed potential for the interchangeability between PET/CT and PET/MR, based on our results. 

## 5. Conclusions

Most of the histogram, geometric, and texture (GLCM and GLRLM) features were interchangeable between PET/CT and PET/MR of patients with brain or head and neck cancers. The wavelet features might be cautiously used with different PET modalities in the image dataset. Further studies are warranted to correct variations of ^18^F-FBPA radiomics that result in different PET modalities.

## Figures and Tables

**Figure 1 biomolecules-11-01659-f001:**
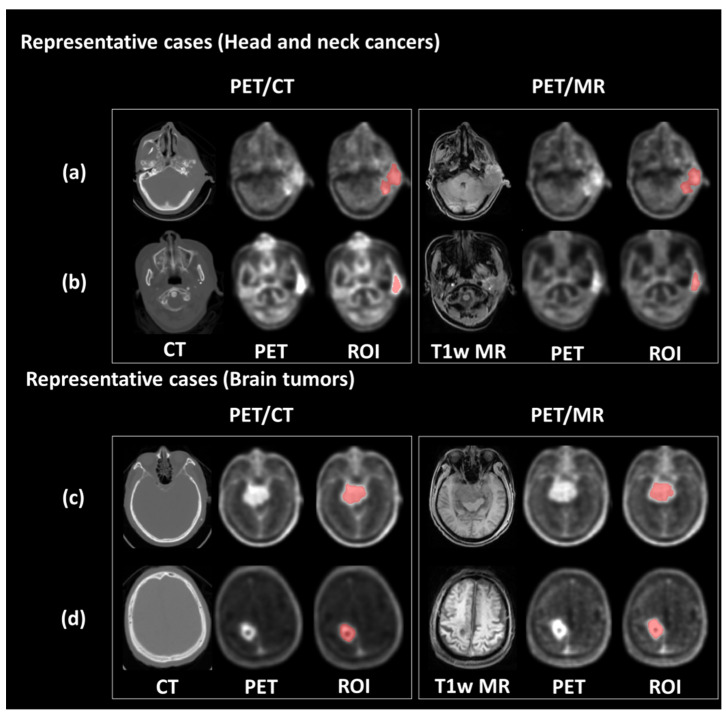
Representative PET/CT and PET/MR images. The upper panel shows the PET images and tumor contour of patients with ear sarcoma (**a**) and tongue cancer (**b**), respectively. The bottom panel shows the PET images and tumor contour of patients with meningioma (**c**) and glioblastoma (**d**), respectively.

**Figure 2 biomolecules-11-01659-f002:**
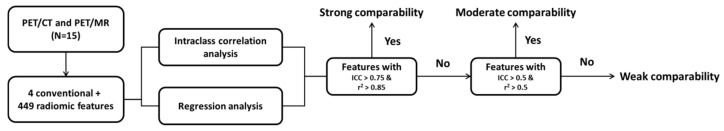
Flowchart of data analysis.

**Figure 3 biomolecules-11-01659-f003:**
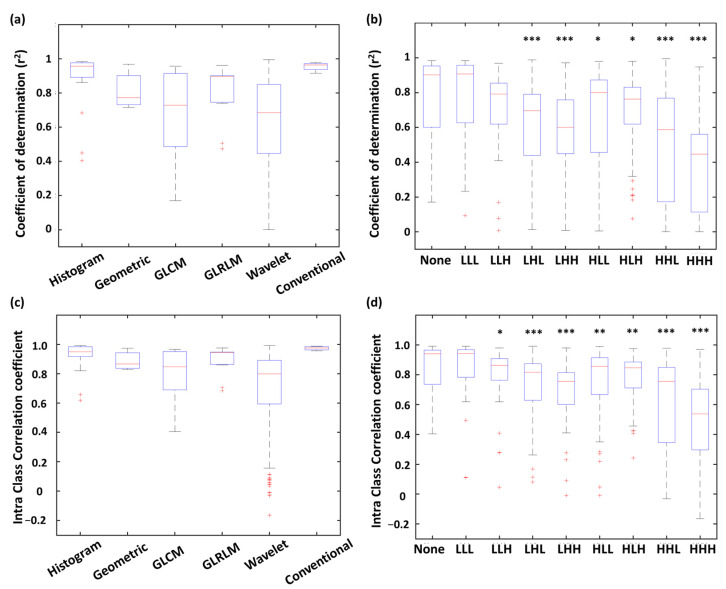
Distribution of r^2^ (**a**,**b**) and ICC *(***c**,**d***)* values of PET/CT and PET/MR comparison among radiomic types and wavelet types. *** *p* < 0.001, ** *p* < 0.01, * *p* < 0.05.

**Figure 4 biomolecules-11-01659-f004:**
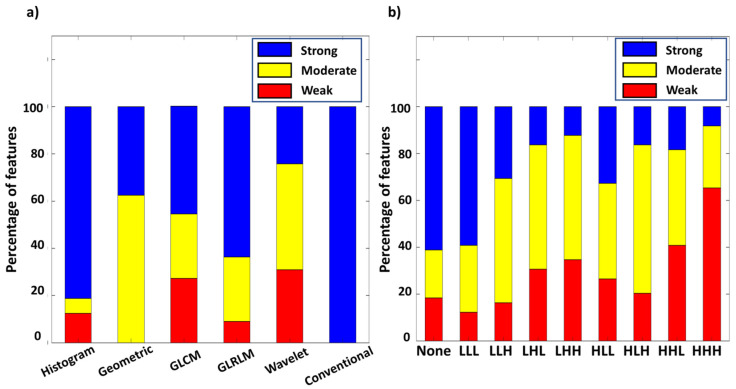
Percentage of different level of comparability among radiomic types (**a**), and wavelet types (**b**).

**Figure 5 biomolecules-11-01659-f005:**
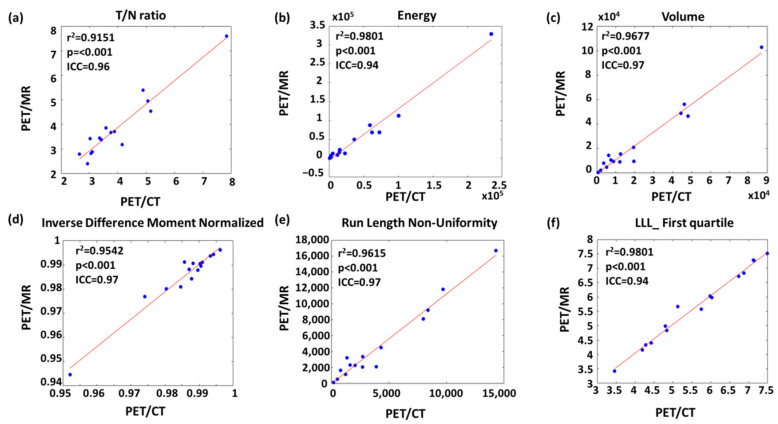
Linear regression of representative (**a**) conventional, (**b**) histogram, (**c**) geometric, (**d**) GLCM-texture, (**e**) GLRLM-texture and (**f**) wavelet features with strong comparability between PET/CT and PET/MR.

**Table 1 biomolecules-11-01659-t001:** Clinical characteristics of study cohort.

Characteristic	Value	Percentage or Range
**Patients (N = 15)**		
Average age	55.4	13–88
Gender (Male:Female)	8:7	
**Tumor types**		
Head and neck cancer		
Ear sarcoma	1	6.7%
Mandible osteosarcoma	1	6.7%
Tongue cancer	1	6.7%
Brain tumor		
Glioblastoma	4	26.4%
Glioma	1	6.7%
Diffuse intrinsic pontine glioma	1	6.7%
Brain metastasis from lung	1	6.7%
Oligoastrocytoma	1	6.7%
Oligodendroglioma	1	6.7%
Astrocytoma	2	13.3%
Meningioma	1	6.7%

## Data Availability

All data generated or analyzed during this study are included in this published article and its Appendix A. To protect the patient privacy, the raw images and data collected in this study can be only accessed by contacting the corresponding author (C.-F.L.).

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
