# Peer review of "Comparison of Conventional and Radiomic Features between 18F-FBPA PET/CT and PET/MR"

_biomolecules, 2021, doi:10.3390/biom11111659_

Round 1
Reviewer 1 Report
It is a very interesting article.
Some suggestions:
1) I encourage authors to add some emblematic figures with captions of brain or head and neck lesions in 18F-FBPA PET/CT and PET/MR of the same patient;
2) in line 261 "tumor tumor" is not correct;
3) it is important to follow scrupulously manuscript preparation instructions and to correct any discrepancies (for example in references).
of para-aortic limph node involvement and colon involvement at CT and PET/CT.
Reviewer 2 Report
Thank you for the paper. I have a few comments.
- Previous studies comparing FDG PET/CT and PET/MRI have shown similar results. How does this study add new informaton?
- You compare sarcoma, an adenocarcinoma and astrocytomas - tumors with great biological differences. Is that methodologically correct?
- The paper has many grammatic errors and needs to be checked again (especially many changes from presence to past tence).
Reviewer 3 Report
The paper written by Liao et al. is an interesting paper that compared 18F-FBPA PET/CT and PET/MR measurements of some conventional and radiomic features.
The idea of the authors is brilliant because the increasing diffusion of PET/MR scanners and the trend to use radiomic features is huge. Thus, the comparison of the main features can have a strong clinical impact and try to ansewer to a question not yet understood.
Of course the number of patients evaluated is relatively low (n 15) but it is impossible to ask a higher number fot his kinf of study.
I believe that the manuscript needs to be accepted after some revisions.
- I don't like the term "routine", it is too vague and not appropriate. I suggest conventional or classic as alternative.
- Paragraph 2.1 and 2.2 have the same title (Patient cohort), is it normal?
- in the introduction about he strengths and limitations of PET/CT and PET/MR I suggest add the higha vailability of PET/CT tomogaph compared to PET/MRI, the strong and long experience in PET/CT, and the need of radiologist with skills in MRI (in not all countries a nuclear medicine physician is also a radiologist)
- who performed radiomic analysis? please add initials. Is it an expert about images analysis?
- in the table 1 please add average before age and specify the last two types of brain tumors
- discussion is too long and heavy, I mean that some parts could be reduced without losting the concept. Please modify
Round 2
Reviewer 2 Report
Thank you for the revised manuscript.
I have no further comments.